# Adaptive patch foraging in deep reinforcement learning agents

**Nathan J. Wispinski**                                     *nathan3@ualberta.ca*
*University of Alberta, Edmonton, Canada*
*(work conducted while an intern with DeepMind, Edmonton, Canada)*

**Andrew Butcher**                                         *abutcher@deepmind.com*
*DeepMind, Edmonton, Canada*

**Kory W. Mathewson**                                     *korymath@deepmind.com*
*DeepMind, Montreal, Canada*

**Craig S. Chapman**                                       *c.s.chapman@ualberta.ca*
*University of Alberta, Edmonton, Canada*

**Matthew M. Botvinick**                                   *botvinick@deepmind.com*
*DeepMind, London, UK*

**Patrick M. Pilarski**                                    *pilarski@ualberta.ca*
*DeepMind, Edmonton, Canada*
*University of Alberta, Edmonton, Canada*
*Alberta Machine Intelligence Institute (Amii), Edmonton, Canada*

**Reviewed on OpenReview:** *https://openreview.net/forum?id=a0T3nOP9sB*

## Abstract

Patch foraging is one of the most heavily studied behavioral optimization challenges in biology. However, despite its importance to biological intelligence, this behavioral optimization problem is understudied in artificial intelligence research. Patch foraging is especially amenable to study given that it has a known optimal solution, which may be difficult to discover given current techniques in deep reinforcement learning. Here, we investigate deep reinforcement learning agents in an ecological patch foraging task. For the first time, we show that machine learning agents can learn to patch forage adaptively in patterns similar to biological foragers, and approach optimal patch foraging behavior when accounting for temporal discounting. Finally, we show emergent internal dynamics in these agents that resemble single-cell recordings from foraging non-human primates, which complements experimental and theoretical work on the neural mechanisms of biological foraging. This work suggests that agents interacting in complex environments with ecologically valid pressures arrive at common solutions, suggesting the emergence of foundational computations behind adaptive, intelligent behavior in both biological and artificial agents.

## 1 Introduction

Patch foraging is one of the most critical behavioral optimization problems that biological agents encounter in nature. Almost all animals forage, and must do so effectively to survive. In patch foraging theory, spatial patches are frequently modeled as exponentially decaying in resources, with areas outside of patches as having no resources (Charnov, 1976). Agents are faced with a decision about when to cease foraging in a depleting patch in order to begin travelling some distance to a richer patch. Research has shown that animals are adaptive patch foragers in this context, intelligently staying in patches for longer when the environment is

resource-scarce, and staying a shorter time in patches when the environment is resource-rich (Cowie, 1977; Hayden et al., 2011; Kacelnik, 1984; Krebs et al., 1974).

Despite its importance to biological intelligence, patch foraging is understudied in artificial intelligence research. Computational models of patch foraging are often agent-based models with fixed decision rules (Pleasants, 1989; Tang & Bennett, 2010), although recent work has involved the use of tabular reinforcement learning models (Constantino & Daw, 2015; Goldshtein et al., 2020; Miller et al., 2017; Morimoto, 2019). Additionally, neural networks trained using methods such as Hebbian learning have displayed foraging behavior in separate ecological tasks such as patch selection (Coleman et al., 2005; Montague et al., 1995; Niv et al., 2002). Many deep reinforcement learning agents are able to successfully search environments for rewarding collectibles like apples while avoiding obstacles and/or enemies (as in gridworlds or popular video games; e.g., Lin, 1991; Mnih et al., 2015; Platanios et al., 2020), and even to stay or switch in the face of decaying rewards (Shuvaev et al., 2020). However, these environments significantly differ from core principles of theoretical and experimental foraging research. Namely, many environments lack the repeating, temporally evolving tradeoff between immediate, decaying resources and delayed, richer ones (Stephens & Krebs, 2019).

Patch foraging is especially feasible to study computationally given that it has a known optimal solution—the marginal value theorem (MVT; Charnov, 1976). In short, the MVT states that the optimal solution is to cease foraging within a patch and begin traveling toward a new patch when the reward rate of the current patch drops below the average reward rate of the environment. Foraging is so important to the survival of biological agents that theorists argue that the foraging behavior of animals is not only adaptive, but approaches this MVT optimal behavior in natural environments because of strong selective pressures (Charnov & Orians, 2006; Pearson et al., 2014; Stephens & Krebs, 2019). Many animals, including humans, have been shown to behave optimally in patch foraging tasks in the wild and in the laboratory. For example, human mushroom foragers (Pacheco-Cobos et al., 2019), rodents (Lottem et al., 2018; Vertechi et al., 2020), and birds, fish, and bees (Cowie, 1977; Krebs et al., 1974; Stephens & Krebs, 2019) all behave consistent with the MVT solution of optimal patch foraging (although many examples of suboptimal patch over-staying exist; see Nonacs, 2001).

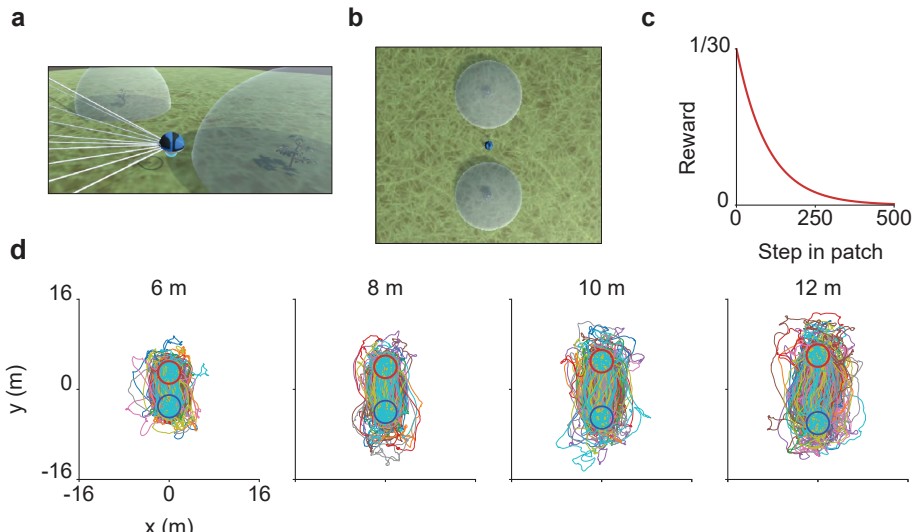

Figure 1: Task. **a)** Mock-up of the 3D foraging environment and agent with LIDAR rays. **b)** Overhead view. An agent starts each episode between two equidistant patches. **c)** The agent receives exponentially decreasing reward on every step it is within a patch. When the agent enters one patch, the opposite patch is immediately refreshed to its starting reward state. **d)** Overhead spatial trajectories of a representative trained agent in each evaluation environment.

However, the optimal patch foraging solution may be difficult to discover using many frequently used deep reinforcement learning approaches. The MVT dictates a comparison between the long-run average reward rate of the environment with the instantaneous reward rate of the current patch. This value comparison requires multiscale temporal resolutions, both local and global, which can be difficult to represent using model-free reinforcement learning, but are seen in animal cognition and neural dynamics (Badman et al., 2020). Further, model-free reinforcement learning potentially requires significant temporal exploration involved with the trial-and-error learning of leaving patches at different depletion times. Finally, changes in the resource-richness of the environment or the agent's movement time between patches impact the optimal patch time, as prescribed by the MVT.

Given the potential difficulty for deep reinforcement learning agents to learn to patch forage, how do biological agents solve the same problem? Theoretical work suggests that a simple evidence accumulation mechanism, similar to one commonly proposed for perceptual and value-based decision making in mammals (Brunton et al., 2013; Gold & Shadlen, 2007; Hanks & Summerfield, 2017; Wispinski et al., 2020), can approximate MVT solutions in complex environments (Davidson & El Hady, 2019). Neural recordings from non-human primate cingulate cortex suggests that such a mechanism may underlie decisions in a computerized patch foraging task (Hayden et al., 2011). Past work in deep reinforcement learning has used task paradigms from animal research to probe the abilities of agents to learn abstract rule structures (Wang et al., 2016), detour problems (Banino et al., 2018), and perceptual decision making (Song et al., 2017). In these studies, internal dynamics show patterns similar to those recorded from animals completing the same behavioral task, suggesting the discovery of similar mechanisms solely through reward learning (Silver et al., 2021). Further, studying animal intelligence, the behavioral pressures that shaped biological intelligence (Cisek, 2019; Stephens & Krebs, 2019), and their neural implementation is likely to expedite progress in artificial intelligence (Hassabis et al., 2017).

Here, we first ask if deep reinforcement learning agents can learn to forage in a 3D patch foraging environment inspired by experiments from behavioral ecology. Second, we ask whether these agents forage intelligently—adapting their behavior to the environment in which they find themselves. Next, we investigate if agent foraging behavior in these environments approaches the known optimal solution. Finally, we investigate how these agents solve the patch foraging problem by interrogating internal dynamics in comparison to theory and neural dynamics recorded from non-human primates.

This paper offers a number of novel contributions. We demonstrate:

1. The first investigation of deep recurrent reinforcement learning agents in a complex ecological patch foraging task;

2. Deep reinforcement learning agents that learn to adaptively trade off travel time and patch reward in patterns similar to biological foragers;

3. That these agents approach optimal patch foraging behavior when accounting for temporal discounting;

4. That the internal dynamics of these agents show key patterns that are similar to neural recordings from foraging non-human primates and patterns predicted by foraging theory.

This paper is an empirical investigation into the emergence of complex patch foraging behavior, and offers a model to the biology community with which to study patch foraging. Additionally, these results add to the neuroscientific literature on how biological agents approximate the optimal solution using a general decision mechanism, and how adaptive behavior arises from simple changes in this mechanism.

## 2 Experiments

### 2.1 Environment

A continuous 3D environment was selected to approximate the rich sensorimotor experience involved in ecological foraging experiments. The environment consisted of a 32 x 32 m flat world with two patches (i.e., half spheres) equidistant from the center of the world (Figure 1a; see Cultural General Intelligence

Team et al., 2022). Patches always had a diameter of 4 m. Agents started each episode at the middle of the world, facing perpendicular to the direction of the patches. Each episode terminated after 3600 steps. Agents received a reward of zero on each step they were outside of both patches. When an agent was within a patch, it received reward according to the exponentially decaying function, $r(n) = N_0 e^{-\lambda n}$, where $n$ is the number of non-consecutive steps the agent has been inside a patch without being inside the alternative patch. In this way, as soon as an agent entered a patch, the alternative patch was refreshed to its initial reward state (i.e., $n = 0$). As such, agents are faced with a decision about how long to deplete the current patch before traveling toward a newly refreshed patch. For all experiments, the initial patch reward, $N_0$, was set to 1/30, and the patch reward decay rate, $\lambda$, was set to 0.01 (Figure 1c). The surface color of each patch changed proportional to the reward state of the patch in RGB space. Patches changed color from white (i.e., [1, 1, 1]) to black (i.e., [0, 0, 0]) following the function, $r(n)/N_0$. In this way, agents had access to the instantaneous reward rate of the patch through patch color, rather than having to estimate patch reward rate by estimating the decay function and keeping track of steps spent within a patch. We chose to make instantaneous patch reward information available in the environment as biological agents often have complete or partial sensory information regarding the current reward state of a patch (e.g., visual input of apple density on a tree).

## 2.2 Agents

The agents observe the environment through a LIDAR sensor-based observation sensor. LIDAR-based sensing is common for physical robots (Malavazi et al., 2018), and agents in other simulated environments (e.g., Baker et al., 2019; Cultural General Intelligence Team et al., 2022). The LIDAR sensor casts uniformly across 3 rows of 8 rays. Rays are evenly spaced, with azimuth ranging from -45° to +45°, and altitude ranging from -30° to +30°. Each ray returns an encoding of the first object it intersects with which includes object type, color, and distance of the object from the agent. Object type is encoded as a one-hot vector for each valid object type (ground plane, patch sphere, no-intersection; e.g., [0, 1, 0]). Color is encoded for patch spheres only, as [R,G,B] where each is a float [0, 1]. Distance is encoded as a float [0, 1] normalized to the maximum LIDAR distance (128 m). This corresponds to a LIDAR space with 24 channels. Agents analyzed in the dynamics results section below instead had 14 rows of 14 rays with azimuth ranging from 0° to 360°, and altitude ranging from -90° to +90°, consistent with (Cultural General Intelligence Team et al., 2022). LIDAR differences did not substantially impact agent behavior. LIDAR inputs were convolved (24 output channels, 2x2 kernel shape), before they were concatenated with the reward and action taken on the previous step. These values were then passed through a MLP (3 layers of 128, 256, and 256 units) and a LSTM layer (256 units). Finally, LSTM outputs were passed to an actor and a critic network head.

The action space is 5-dimensional and continuous (adapted from Cultural General Intelligence Team et al., 2022). Each action dimension takes a value in [−1, 1]. The dimensions correspond to 1) moving forward and backwards, 2) moving left and right, 3) rotating left and right, 4) rotating up and down, and 5) jumping or crouching. Agents can take any combination of actions simultaneously. Movement dynamics are subject to

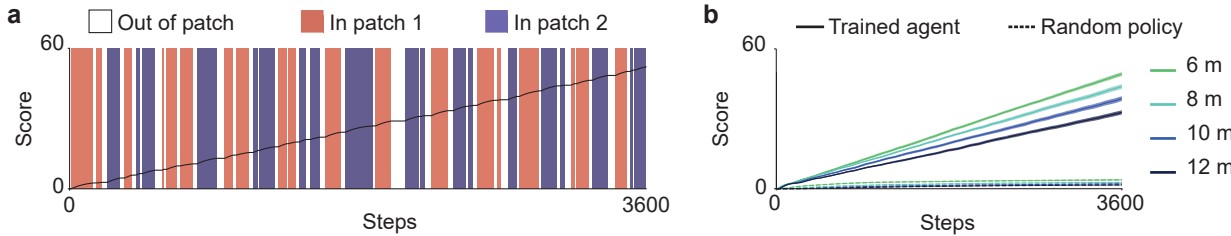

Figure 2: Performance. **a)** Agent behavior from a representative evaluation episode. Shaded regions define when the agent is outside of any patch (white), inside patch 1 (red), or inside patch 2 (blue). **b)** Episode score for a representative trained agent (solid lines), and a representative agent with a random uniform action policy (dashed lines) in each evaluation environment. Shaded regions denote standard errors across evaluation episodes.

the inertia of the environment. Actions were taken by sampling from Gaussians parameterized by the policy network head output for each action dimension.

We use a state-of-the-art continuous control deep reinforcement learning (RL) algorithm for training our agents: maximum a posteriori policy optimization (MPO; Abdolmaleki et al., 2018). MPO is an actor-critic, model-free RL algorithm which leverages samples to compare different actions in a particular state and then updates the policy to ensure that better actions have a high probability of being sampled. MPO alternates between policy improvement which updates the policy $\pi$ using a fixed Q-function and policy evaluation which updates the estimate of the Q-function. Following previously described methods, agents were trained in a distributed manner with each agent interacting with 16 environments in parallel (Cultural General Intelligence Team et al., 2022). Agent architecture and training hyperparameters were the same as in Cultural General Intelligence Team et al. (2022), unless otherwise stated. Experience was saved in an experience buffer. Agent parameter updates were accomplished by sampling batches and using MPO to fit the mean and covariance of the Gaussian distribution over the joint action space (Abdolmaleki et al., 2018). Agent architecture, observation space, and the MPO algorithm were selected in part because of the success of these implementational choices in more complex environments (see Cultural General Intelligence Team, 2022).

Three agents were trained in each of four discount rate treatments ($N = 12$), selected on the basis of MVT simulations (Figure 3d). Agents were each initialized with a different random seed, and trained for $12e^7$ steps using the Adam optimizer (Kingma & Ba, 2014) and a learning rate of $3e^{-4}$. On each training episode, patch distance was drawn from a random uniform distribution between 5 m and 12 m, and held constant for each episode. Trained agents were evaluated on 50 episodes of each evaluation patch distance (i.e., 6, 8, 10, and 12 m). By manipulating patch distance, we vary the amount of time it takes agents to travel between patches, which in turn varies the resource-richness of the environment—similar to many animal experiments on adaptive patch foraging behavior (Cowie, 1977; Kacelnik, 1984). If an agent was within a patch at the end of an evaluation episode (e.g., Figure 2a), this final patch encounter was excluded from all analyses, as no distinct patch leave behavior could be verified to determine the total steps in this patch.

Videos of a representative agent during training and evaluation are available in the supplementary material.

## 3    Results

For several results below, we fit data using a linear regression of the form, $y = bx + a$, where free parameter $b$ is the slope of the fitted line, and $a$ is a constant. $b$ references the predicted relationship, negative or positive, of the rate of change in $y$ (e.g., steps in patch; Figure 3a) per unit change in $x$ (e.g., patch distance in meters). In text, we report $b$, along with its standard error (e.g., $b = -1.50 \pm 0.50$). For full statistical reporting, see Appendix A.1.

### 3.1    Environment adaptation

Trained agents displayed behavior consistent with successful patch foraging—agents learned to leave patches before they were fully depleted of reward (mean leaving step = 121.7), and traveled for several steps without reward in order to reach a refreshed patch (mean travel steps between patches = 57.7; e.g., Figure 2a). We computed each agent's mean final score in each patch distance evaluation environment (e.g., single representative agent in Figure 2b) to analyze whether agent score varied systematically with patch distance. Analysis showed agents achieved a higher score on episodes where patches were closer together ($p < 0.05$, linear regression slope $b = -5.82 \pm 0.21$).

In patch foraging theory and experimental animal research, animals intelligently adapt patch leaving times to the environment in which they find themselves. Theory predicts agents monotonically increase their steps in patch when the distance between patches increases (Charnov, 1976; Stephens & Krebs, 2019). Similarly, in biological data, agents stay a longer time in patches when alternative patches are further away (Cowie, 1977; Hayden et al., 2011; Kacelnik, 1984; Krebs et al., 1974). This adaptive behavior is present in the current agents—trained agents increased their patch leaving times when patch distance increased, leaving patches later when travel distance was higher ($p < 0.05$, $b = 9.60 \pm 0.87$; Figure 3a). These results support

our contribution that the current deep reinforcement learning agents learn to adaptively trade off travel time and patch reward in patterns similar to biological foragers.

## 3.2 Optimality

Above we show that trained agents are able to successfully forage, and intelligently adapt their foraging behavior to the environment in accordance with patch foraging theory (Charnov, 1976; Stephens & Krebs, 2019), and animal behavior (e.g., Cowie, 1977). However, do these agents adapt optimally according to the marginal value theorem (MVT), like many animals (Stephens & Krebs, 2019)? As stated above, the MVT provides a simple rule for when to leave patches optimally (Charnov, 1976; Shuvaev et al., 2020). That is, an agent should leave a patch when the reward rate of the patch drops below the average reward rate of the environment (Figure 3c).

The current agents however use temporal discounting methods, which exponentially diminish rewards in the future. In other words, agents are tasked with solving a *discounted* patch foraging problem. Given that agents are effectively asked to compare the values between the current patch reward on the next step relative to a refreshed patch reward after several travel steps, temporal discounting encourages longer patch leaving times relative to predictions from the MVT. Below, we evaluate agents relative to the MVT, which animals are typically compared against (Cowie, 1977; Charnov & Parker, 1995; Hayden et al., 2011; Krebs et al., 1974; Pacheco-Cobos et al., 2019; Stephens & Krebs, 2019). We also attempt to account for temporal discounting in the MVT to compare agents against the task that they are given.

For each agent and evaluation environment (e.g., 6 m), we can estimate the average reward rate of the environment by calculating the average reward per step for each evaluation episode. Over all evaluation episodes for each agent and environment, this provides an estimate of the optimal patch leaving step (see Figure 3c and Figure 3g). Here, we take each agent's mean patch leaving time relative to the MVT across patch distance environments, and test if these 12 values (one for each agent) are significantly different from

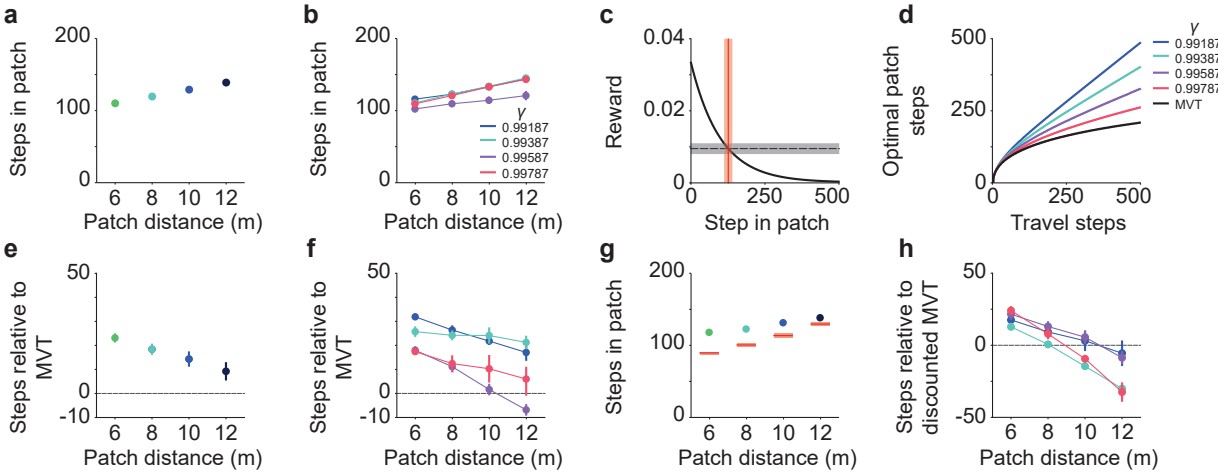

Figure 3: Patch leaving times. **a)** Average of all agents. **b)** Average of agents grouped by discount rate. **c)** Graphical model of the MVT solution. Where the patch reward rate (solid black line) intersects the observed average reward rate of the environment as determined by the agent's behavior (dashed horizontal black line), determines the MVT optimal average patch leaving time (solid vertical red line). **d)** Patch leaving times prescribed by the MVT (black), and simulation results for the MVT considering discount rates. **e)** Mean difference between the observed and optimal patch leaving time for all agents, and **f)** agents grouped by discount rate. **g)** Representative single trained agent patch leaving times (dots) against the MVT solution (red lines). **h)** Mean difference between the observed and discounted MVT patch leaving time for agents grouped by discount rate. All vertical lines and/or shaded regions denote standard errors over agent means in each patch distance evaluation environment.

zero (Figure 3e) using a one-sample t-test. Comparing the difference between average observed and MVT optimal patch leaving times, agents tend to overstay in patches relative to the MVT optimal solution ($p < 0.05$; mean = 15.8 steps above MVT optimal, standard error = 2.7; Figure 3e). These results are consistent with the patch foraging behavior of humans and other animals in computerized laboratory tasks (Cash-Padgett & Hayden, 2020; Constantino & Daw, 2015; Hutchinson et al., 2008; Kane et al., 2017; Nonacs, 2001), and are also consistent with temporal discounting predictions.

We now ask if agents trained with higher temporal discounting rates behave closer to MVT optimal (Figure 3f). Here, we take the mean patch leaving time relative to the MVT across patch distance environments, and group these values by agent discount rate, leaving 3 agents per discount rate. Using linear regression, we test whether the difference between observed and MVT optimal patch leaving times decreases with discount rate. As expected, agents trained with higher temporal discounting rates tend to behave closer to MVT optimal ($p < 0.05$, $b = -2784.01 \pm 992.19$; Figure 3f; for details see Appendix A.1.2).

Are agents then optimal after accounting for temporal discounting rates in the MVT solution? We accounted for the temporal discounting rate by simulating individual stay and leave decisions at many patch leaving steps (for details, see Appendix A.2). Agents could either stay for an additional step of reward before leaving a patch, or immediately leave the patch, where the subsequent 5000 steps were simulated as alternating between a fixed number of steps in a patch and a fixed number of steps traveling between patches. For example, on step 42 within a patch, and given a future fixed travel time of 50 steps and a future fixed patch time of 100 steps, is it more beneficial to leave immediately or stay in the patch for an additional step of reward? Over a grid of subsequent fixed patch and travel steps, the difference in the discounted return (sum of discounted rewards) between each stay/leave decision provided an indifference curve, where the 5000-step discounted return was equal for staying relative to leaving. Where this stay/leave indifference step matched the fixed patch steps provided an approximation of an average patch time where the value of leaving is about to exceed the value of staying.

After accounting for each agent's temporal discounting rate in the MVT (Figure 3d), we ask if agent patch leaving times approach the optimal solution (Figure 3h). Similar to above, we take each agent's mean patch leaving time relative to the discounted MVT solution across patch distance environments, and test if these 12 values are significantly different from zero (Figure 3h) using a one-sample t-test. Comparing the difference between average observed and discounted MVT optimal patch leaving times, agents were not significantly different from the optimal solution ($p = 0.74$; mean = 0.9 steps above discounted MVT optimal, standard error = 2.8; Figure 3h)—although agents appear to slightly understay or overstay in patches depending on the environment (see Appendix A.1.2). Overall, these results support our contribution that the current agents approach optimal patch foraging behavior when accounting for temporal discounting.

### 3.3 Dynamics and patch leaving time variability

Above we show that trained agents approach optimal patch leaving time behavior when accounting for temporal discounting. Here, we investigate how these agents decide to leave a patch by interrogating internal dynamics in comparison to theory and neural dynamics recorded from non-human primates.

Theory and modeling work have shown that patch foraging decisions can be made using a simple evidence accumulation mechanism, which can approximate MVT solutions in complex environments (Davidson & El Hady, 2019). Evidence accumulation is a general decision making mechanism, which has been proposed to underlie many perceptual- and value-based decisions in animals (Brunton et al., 2013; Gold & Shadlen, 2007; Wispinski et al., 2020). In brief, a decision variable representing the degree of evidence in support of a decision (i.e., leaving the current patch) accumulates over multiple time steps until it reaches a threshold (Figure 4a). When a decision variable reaches threshold dictates when a decision is made (i.e., when the agent commits to leaving a patch). Evidence accumulation is especially useful when considering evidence that is delivered over multiple time points (e.g., rewards in a patch, Davidson & El Hady, 2019; Hayden et al., 2011; or noisy visual input, Gold & Shadlen, 2007; Ratcliff et al., 2016; Wispinski et al., 2020). Key mechanisms in this model typically include the slope with which evidence accumulates, and the distance that evidence needs to travel from baseline to threshold for a decision to be made (Gold & Shadlen, 2007; Ratcliff et al., 2016). For example, when evidence for a decision is stronger, the decision variable tends to

accumulate with a higher slope, reaching threshold sooner (Figure 4a). Similarly, when a decision requires less evidence, the distance between baseline and threshold can be decreased so decisions are made sooner. Both of these mechanisms are typically subject to variability in the internal state of the decision maker (Gold & Shadlen, 2007; Ratcliff et al., 2016).

Neural recordings in primate cingulate cortex similar to an evidence accumulation mechanism have been observed during a computerized patch foraging task, suggesting that such a mechanism may underlie biological solutions to the patch foraging problem (Hayden et al., 2011). Here we investigate LSTM layer activity in a single trained agent, and show several emergent similarities with neural data in foraging primates. The results described below are consistent with several other agents we investigated.

In this section we ask if variability in the internal state of the agent can explain earlier or later patch leaving times within the same patch distance environment, as in biological agents (Hayden et al., 2011). We took patch encounters where an agent had first entered a newly refreshed patch until it first left that patch. In these encounters, agents experience the same exponential decrease in reward within a patch but displayed variable patch leaving times. As in Hayden et al. (2011), we divided these data into quartiles based on patch leaving times. Patch encounters were split into exclusive groups named: Earliest (25th percentile), Early

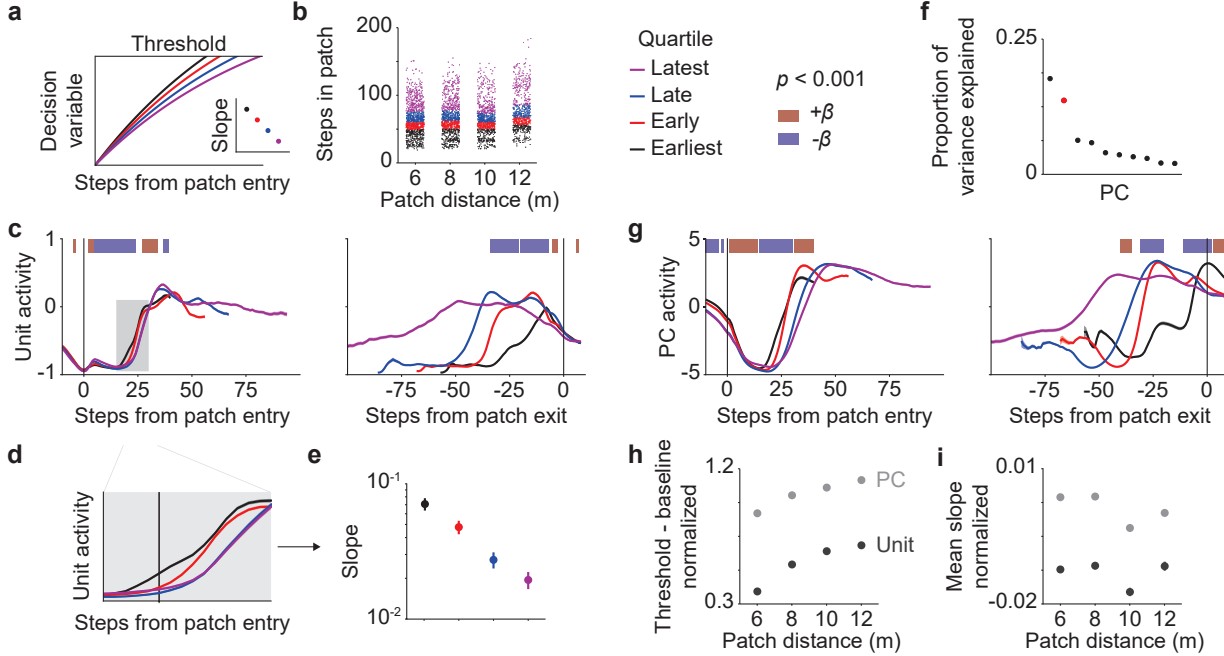

Figure 4: Dynamics from a single trained agent. **a)** Example of an idealized evidence accumulation process. A decision variable accumulates to a threshold, which determines patch leaving time. Here, variability in the slope of the decision variable explains patch leaving time differences. **b)** Patch encounters for a single agent split into patch leaving time quartiles in each evaluation environment. **c)** Mean activity of an example LSTM unit aligned to patch entry (left) and patch exit (right), for each quartile of patch leaving times. Shaded regions along activity traces denote standard errors across patch encounters. Shaded bars at the top of the plot indicate steps where there is a significant slope-quartile relationship (negative in blue, positive in red). For patch entry-aligned data, traces in each quartile are plotted until median patch leaving time. **d)** Zoomed-in region of **c**. **e)** Mean slope by patch leaving time quartile for step 20 in **d** (log-scale). **f)** Principal components decomposition of LSTM layer dynamics. Selected principal component (PC) in red. **g)** Activation of the selected principal component. **h)** Mean difference in example unit and example PC activity from patch exit to patch entry in each evaluation environment. **i)** Mean slope of example unit and example PC in each evaluation environment. All vertical lines and/or shaded regions denote standard errors across patch encounters.

(50th percentile), Late (75th percentile), and Latest (100th percentile). These groups were taken equally from each evaluation environment (6, 8, 10, 12 m) so results were independent of patch distance (Figure 4b).

Using these groups, we can investigate if earlier patch leaving times in each evaluation environment had a higher slope of rising activity, as predicted by theory and biological data (Figure 4a). We perform a linear regression of the slope of LSTM unit change against patch leaving time quartile (e.g., Figure 4e) at every time step, using data from all individual patch encounters. We performed these sliding regressions on both patch entry-aligned data (left panel Figure 4c), and patch exit-aligned data (right panel Figure 4c). Primate data show that cingulate neurons have a higher slope of activity before the Earliest patch leaving times, and a lower slope of activity before the Latest patch leaving times (inset of Figure 4a). Here we find the same pattern in a number of LSTM units for several steps after patch entry, but before patch exit (blue shaded bars; left panel of Figure 4c). In the example unit shown, there are 19 consecutive steps where there is a significantly higher slope for encounters that had shorter patch leaving times (e.g., step 20: $p < 0.001$, $b = -0.0174 \pm 0.002$; Figure 4e). In other words, the pattern of results in Figure 4e match the idealized results consistent with primate neural data and theory in the inset of Figure 4a. Additionally, only 1/10 steps had a significant relationship just before and after the patch encounter.

A principal components dimensionality reduction was performed on the LSTM data, and most results for the example unit can also be seen in a component accounting for roughly 14% of LSTM layer variability (Figure 4f). The expected negative relationship between activity slope and patch leaving quartile (e.g., Figure 4a and Figure 4e) was statistically significant for 15 consecutive steps after patch entry (blue shaded bars; left panel of Figure 4g).

Additionally, activity traces appear to begin at a similar level at patch entry, and appear to converge just before patch exit (Figure 4c and g), suggesting that changes in the distance activity must travel from baseline to threshold do not underlie patch leaving time variability (although not significantly; see Appendix A.1). Overall, this suggests that variability in the slope of rising activity can explain variability in the agent's patch leaving times, similar to primate neural recordings during patch foraging (Hayden et al., 2011). These results support the first half of our contribution that internal dynamics of these agents show key patterns that are similar to neural recordings from foraging non-human primates and patterns predicted by foraging theory.

### 3.4 Dynamics and environment adaptation

Above we show that within a patch distance environment, variability in the slope of LSTM layer activity predicts patch leaving time. We now turn to investigate the same relationships, but *between* patch distance environments. The behavioral analysis above showed that agents adapt patch leaving times to the resource-richness of the environment (Figure 3a). Agents stayed in patches for more steps when travel time to a new patch was longer, and stayed in patches for fewer steps when travel time to a new patch was shorter.

Under an evidence accumulation mechanism, patch leaving times can be adapted by changing the slope of activity, or by changing the distance a decision variable needs to accumulate from baseline to threshold in each evaluation environment. Primate neural data suggests that when travel times between patches increase, the average slope of neural activity decreases, and the distance between baseline and threshold increases—both acting to prolong the time the animals stay in a patch (Hayden et al., 2011). We investigated which of these changes, if any, may drive the increase of patch leaving time when patches are further apart.

Decreasing the slope with which a decision variable accumulates toward a threshold prolongs patch leaving times. Here we took the average slope of activity during significant time steps (i.e., blue bars; Figure 4c and g) for every patch encounter, and separated these data by patch distance environment. Using a linear regression, we find no significant relationship between the average slope of activity and patch distance environment in the example unit ($p = 0.18$, $b = -0.00026 \pm 0.00020$; Figure 4i), nor for the selected PC ($p = 0.13$, $b = -0.0034 \pm 0.002$; Figure 4i).

Increasing the distance a decision variable needs to travel from baseline to threshold is another mechanism which prolongs patch leaving times (Davidson & El Hady, 2019). Here we took the difference between activity at patch exit and activity at patch entry for every patch encounter, and separated these data by

patch distance. This difference in activity approximates the range a decision variable may need to span to complete a patch leaving decision (e.g., Figure 4a). Using a linear regression, we find a positive relationship between activity range and patch distance environment for both the example unit ($p < 0.05$, $b = 0.053 \pm 0.003$; Figure 4h), and for the selected PC ($p < 0.05$, $b = 0.20 \pm 0.020$; Figure 4h).

Through an evidence accumulation framework, these results suggest that adaptive behavior across environments is accomplished via changes in the distance a decision variable must travel between baseline and threshold. In other words, when it takes less time to travel to a new patch, distance between baseline and threshold decrease, shortening the agent's time in the current patch. When it takes more time to travel to a new patch, distance between baseline and threshold increase, prolonging the agent's time in the current patch. This mechanism is similar to work which suggests that decision thresholds during foraging may be estimated by an exponential moving average of past rewards (Constantino & Daw, 2015; Davidson & El Hady, 2019; Shuvaev et al., 2020) to approximate the average reward rate of the environment in the MVT (Charnov, 1976). In contrast, primate neural recordings suggest that adaptation between environments with short or long travel times between patches is also driven by changes in the slope of activity (Hayden et al., 2011)—a different mechanism within an evidence accumulation framework, albeit with very similar behavioral results in this context. These results support the second half of our contribution that internal dynamics of these agents show key patterns that are similar to neural recordings from foraging non-human primates and patterns predicted by foraging theory—although in this case results are different but are consistent with the same underlying mechanism.

## 4  Discussion

Here we tested deep reinforcement learning agents in a foundational decision problem facing biological agents—patch foraging. We find that these agents successfully learn to forage in a 3D patch foraging environment. Further, these agents intelligently adapt their foraging behavior to the resource-richness of the environment in a pattern similar to many biological agents (Cowie, 1977; Hayden et al., 2011; Kacelnik, 1984; Krebs et al., 1974; Stephens & Krebs, 2019).

Many animals (Cowie, 1977; Krebs et al., 1974; Stephens & Krebs, 2019), including humans in the wild (Pacheco-Cobos et al., 2019), have been shown to be optimal patch foragers. The deep reinforcement learning agents investigated here also approach optimal foraging behavior after accounting for discount rate. These results are similar to those from humans and non-human primates in computerized patch foraging tasks—participants tended to overstay relative to the MVT solution, but this discrepancy was significantly reduced after accounting for discount rate or risk sensitivity (Constantino & Daw, 2015; Cash-Padgett & Hayden, 2020). Together, these results raise questions as to how and why humans seemingly discount future rewards in lab foraging (Constantino & Daw, 2015; Hutchinson et al., 2008), but show undiscounted patch leaving times in nature (Pacheco-Cobos et al., 2019). Although outside the scope of this paper, these issues may be reconciled with further work on foraging using different reinforcement learning approaches, such as average reward reinforcement learning (Kolling & Akam, 2017; Sutton & Barto, 2018), especially in deep reinforcement learning models (Shuvaev et al., 2020; Zhang & Ross, 2021), to potentially better model ecological decision making in biological agents. While the current paper considers only the MPO reinforcement learning algorithm due to its success in a similar environment (Cultural General Intelligence Team et al., 2022), future work may find improvements in learning to solve the patch foraging problem with alternative methods.

In several agents that learned to adaptively patch forage, internal dynamics emerged to resemble several key properties of single-cell neural recordings from foraging non-human primates completing a computerized patch foraging task (Hayden et al., 2011). These results complement experimental and theoretical work on how adaptive patch foraging behavior may be accomplished in biological agents by a general decision making mechanism—evidence accumulation (Davidson & El Hady, 2019; Gold & Shadlen, 2007; Wispinski et al., 2020). These results however do not necessarily mean that the agents investigated here, nor biological agents, use an evidence accumulation mechanism to solve the patch foraging problem (Blanchard & Hayden, 2014; Kane et al., 2021). Other strategies, or variations on accumulation models have also been proposed to underlie foraging behavior (Davidson & El Hady, 2019; Kilpatrick et al., 2021; Cazettes et al., 2022). As

in biological research, decision making frameworks provide one way to interpret and test predictions about behavioral and neural data. Emergent patterns in artificial agents completing similar tasks to biological agents provide concrete predictions for neural data and aid in interpretability (Banino et al., 2018; Hassabis et al., 2017; Song et al., 2017; Wang et al., 2016)—especially in ecological tasks like patch foraging where neural data is often more difficult to interpret relative to in-lab, stationary, computerized tasks (Pearson et al., 2014).

In this paper we demonstrate agents capable of patch foraging in a complex environment using a continuous action space. Future experiments may build on the ecological complexity of the environment and interesting agent behavior may arise in environments where the assumptions and predictions of the MVT start to break down (Davidson & El Hady, 2019; Stephens & Krebs, 2019). For example, biological realism may be increased by adding multiple patches, modeling a biologically realistic patch refresh rate, or adding competitive or collaborative agents to the environment (Bidari et al., 2022). Other biological research argues that there are optimal movement strategies for finding unknown patch locations, and that many animals abide by these movement policies (Calhoun et al., 2014; Cisek, 2019; Sims et al., 2008; Tello-Ramos et al., 2015; Woodgate et al., 2017). Given that the current agents generate continuous and complex movement trajectories (Figure 1d), future work may investigate situations in which these optimal movement policies may emerge in artificial agents. The current movement trajectories may also be improved in future experiments by using alternative RL methods or modifying the sensory inputs available to the agents. Finally, foraging frameworks have been successfully extended to explain other aspects of intelligence, such as visual search (Wolfe et al., 2018), or human memory (Hills et al., 2012), which provide another avenue where reinforcement learning models of foraging may aid in intelligence research.

In conclusion, we have trained deep reinforcement learning agents on a complex patch foraging task and for the first time observed the emergence of adaptive, optimal behavior, and neural dynamics that resembled those of biological agents. This paper contributes a model with which biological and artificial intelligence researchers may further understand patch foraging (Frankenhuis et al., 2019)—a fundamental decision problem that strongly guided the evolution of biological intelligence (Cisek, 2019; Stephens & Krebs, 2019). Such paradigms have been, and may continue to be, critical in the continued development of artificial intelligence (Hassabis et al., 2017; Lindsay, 2021).

**Broader Impact Statement**

First, these results suggest the utility of using deep reinforcement learning agents to model foraging behavior and its underlying neural mechanisms in biological agents, especially in more complex environments where the optimal solution is not known. Second, these model-free agents and the patch foraging problem can provide insights into other problems requiring the consideration of value at multiple temporal resolutions, such as information search and resource management applications. Finally, we argue that patch foraging provides a relevant environment from the biological literature to investigate continual learning in artificial agents, as biological agents must adapt their behavior to changing environment statistics (e.g., when changing locations from a scarce to a plentiful environment, or during seasonal changes in resources).

**Acknowledgments**

We are deeply indebted to our colleagues, including Leslie Acker, Andrew Bolt, Michael Bowling, Dylan Brenneis, Adrian Collister, Elnaz Davoodi, Richard Everett, Arne Olav Hallingstad, Nik Hemmings, Edward Hughes, Michael Johanson, Marlos Machado, Drew Purves, Kimberly Stachenfeld, Richard Sutton, Jane Wang, Alexander Zacherl, and the DeepMind Cultural General Intelligence Team for their support, suggestions, and insight regarding this work. We would also like to thank Ben Hayden and Eric Charnov for insightful comments on an earlier version of the manuscript. This work was funded solely by DeepMind. The authors declare no competing interests.

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

# A  Appendix

## A.1  Statistical reporting

### A.1.1  Environment adaptation

We computed each agent's mean final score in each patch distance evaluation environment (e.g., single representative agent in Figure 2b) to analyze whether agent score varied systematically with patch distance. Agents achieved a higher score on episodes where patches were closer together (Linear mixed effects regression with random agent intercept: $b = -5.82 \pm 0.21$, $p = 3.96 \times 10^{-166}$; Figure 2b). Data consisted of 12 mean final scores (one for each agent) $\times$ 4 patch distances.

Trained agents adapted their patch leaving times to the environment, leaving patches later when travel distance is higher (Linear mixed effects regression with random agent intercept: $b = 9.60 \pm 0.87$, $p = 4.03 \times 10^{-28}$; Figure 3a). Data consisted of 12 mean patch leaving times (one for each agent) $\times$ 4 patch distances.

### A.1.2  Optimality

We take each agent's mean patch leaving time relative to the MVT across patch distance environments, and test if these 12 values are significantly different from zero (Figure 3e) using a one-sample t-test. Comparing the difference between average observed and optimal patch leaving times, agents tend to overstay in patches relative to the optimal solution (One-sample t-test: $t(11) = 5.60$, $p = 1.60 \times 10^{-4}$; mean = 15.8 steps above MVT optimal, standard error = 2.7; Figure 3e). Bonferroni-corrected one-sample t-tests show that agents significantly overstayed relative to the MVT solution in all evaluation environments ($ps < 0.0015$), except for 12 m ($p = 0.047$).

We take the mean patch leaving time relative to the MVT across patch distance environments, and group these values by agent discount rate, leaving 3 agents per discount rate. Using linear regression, we test whether the difference between observed and MVT optimal patch leaving times decreases with discount rate. As expected, agents trained with higher temporal discounting rates tend to behave closer to MVT optimal (Linear regression: $b = -2784.01 \pm 992.19$, $p = 0.019$; Pearson correlation: $r(10) = -0.66$, $p = 0.019$; Figure A.1c). Data consisted of 3 agent's mean difference between observed and MVT patch leaving times (averaged across 4 environments) $\times$ 4 discount rates.

We take each agent's mean patch leaving time relative to the discounted MVT solution across patch distance environments, and test if these 12 values are significantly different from zero (Figure 3h) using a one-sample t-test. Comparing the difference between average observed and discounted MVT optimal patch leaving times, agents were not significantly different from the optimal solution (One-sample t-test: $t(11) = 0.34$, $p = 0.74$; mean = 0.9 steps above discounted MVT optimal, standard error = 2.8; Figure 3h). Bonferroni-corrected one-sample t-tests show that agents significantly overstayed in the 6 and 8 m evaluation environments ($ps < 0.0067$), understayed in the 12 m ($p = 0.0023$), and were not significantly different from optimal in the 10 m environment ($p = 0.30$).

### A.1.3  Dynamics and patch leaving time variability

Trained dynamics agents were evaluated on 30 episodes of each evaluation patch distance (i.e., 6, 8, 10, and 12 m). For the individual trained agent presented in Figure 4, evaluation data were processed into 3790 unique patch encounters (mean = 31.6 unique patch encounters per evaluation episode). While single LSTM units (of 256) were analyzed, a principal components analysis (PCA) was also conducted to visualize patterns that accounted for larger amounts of variability in this layer. Exploratory analysis found that PC2 captured several key effects described in Hayden et al. (2011) while also accounting for 14% of LSTM layer variability (the second most in the network). Other PCs are included in Figure A.2 for completeness.

For single-trial dynamics data aligned to patch entry and patch exit (Figure 4c and g), the slope of LSTM unit or PC activity change across consecutive steps was regressed against patch leaving time quartile for 40 in-patch steps and 10 pre- or post-patch steps. Significance threshold for linear regressions at each dynamics

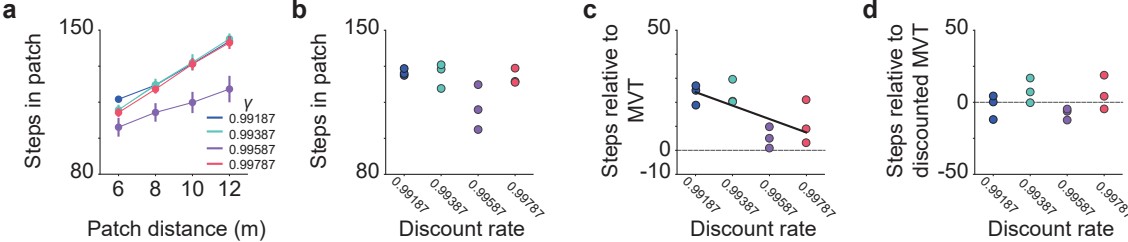

Figure A.1: Patch leaving times. **a)** Agents grouped by discount rate. **b)** Agents grouped by discount rate and collapsed by patch distance environment. **c)** Mean difference between the observed and MVT patch leaving time collapsed by patch distance environment (Linear regression: $b = -2784.01 \pm 992.19$, $p = 0.019$). **d)** Mean difference between the observed and discounted MVT patch leaving time collapsed by patch distance environment. All vertical lines denote standard errors.

time step was Bonferroni-corrected for 50 steps (i.e., p = 0.001) independently for patch entry- and patch exit-aligned data for both the example unit (Figure 4c) and for the example PC (Figure 4g).

We find in several LSTM units a significant relationship between the slope of rising activity and patch leaving time quartile for several steps after patch entry, but before patch exit (blue shaded bars; Figure 4c). In the example unit shown, there are 19 consecutive steps where there is a significantly higher slope for encounters that had shorter patch leaving times ($ps < 0.001$; e.g., step 20 linear regression: $b = -0.0174 \pm 0.002$, $p = 2.36 \times 10^{-14}$; step 20 Pearson correlation: $r(3774) = -0.12$, $p = 2.36 \times 10^{-14}$; Figure 4e). Data for each linear regression consisted of $\sim944$ activity slopes $\times$ 4 patch leaving time quartiles (Earliest, Early, Late, Latest). All results were equivalent when using Pearson correlations instead of linear regressions.

Exit step activity was significantly different between patch leaving time quartiles for the example unit (One-way ANOVA: $F(3, 3775) = 7.15$, $p = 8.76 \times 10^{-5}$). Bonferroni-corrected pairwise follow-up tests show no significant differences other than the mean activity of the Latest patch leaving time quartile from all other conditions ($ps < 0.0012$). For the selected PC, analysis similarly showed exit step activity was significantly different between patch leaving time quartiles (One-way ANOVA: $F(3, 3775) = 320.79$, $p = 1.58 \times 10^{-185}$). Follow-up tests show significant differences between all conditions ($ps < 1.13 \times 10^{-11}$) other than between the Late and Latest patch leaving time quartiles ($p = 0.99$).

### A.1.4   Dynamics and environment adaptation

We took the difference between activity at patch exit and activity at patch entry for every patch encounter, and separated these data by patch distance. We find a positive relationship between activity range and patch distance environment for the example unit (Linear regression: $b = 0.053 \pm 0.003$, $p = 5.88 \times 10^{-56}$; Pearson correlation: $r(3774) = 0.25$, $p = 5.59 \times 10^{-56}$; Figure 4h), and for the selected PC (Linear regression: $b = 0.20 \pm 0.020$, $p = 1.70 \times 10^{-22}$; Pearson correlation: $r(3774) = 0.16$, $p = 1.70 \times 10^{-22}$; Figure 4h). Data consisted of $\sim944$ differences in exit and entry activity $\times$ 4 patch distance evaluation environments (6, 8, 10, and 12 m).

Decreasing the slope with which a decision variable accumulates toward a threshold tends to prolong patch leaving times. For both the example unit and PC, we took the average slope of activity across all steps for which there was a significant predictive relationship between slope and patch leaving time quartile after patch entry (e.g., steps 5 to 23 after patch entry for the example unit; Figure 4c). We find no significant relationship between the average slope of activity and patch distance environment in the example unit (Linear regression: $b = -0.00026 \pm 0.00020$, $p = 0.18$; Pearson correlation: $r(3785) = -0.022$, $p = 0.18$; Figure 4i), nor for the selected PC (Linear regression: $b = -0.0034 \pm 0.002$, $p = 0.13$; Pearson correlation: $r(3784) = -0.025$, $p = 0.13$; Figure 4i). Data consisted of $\sim944$ average activity slopes $\times$ 4 patch distance evaluation environments.

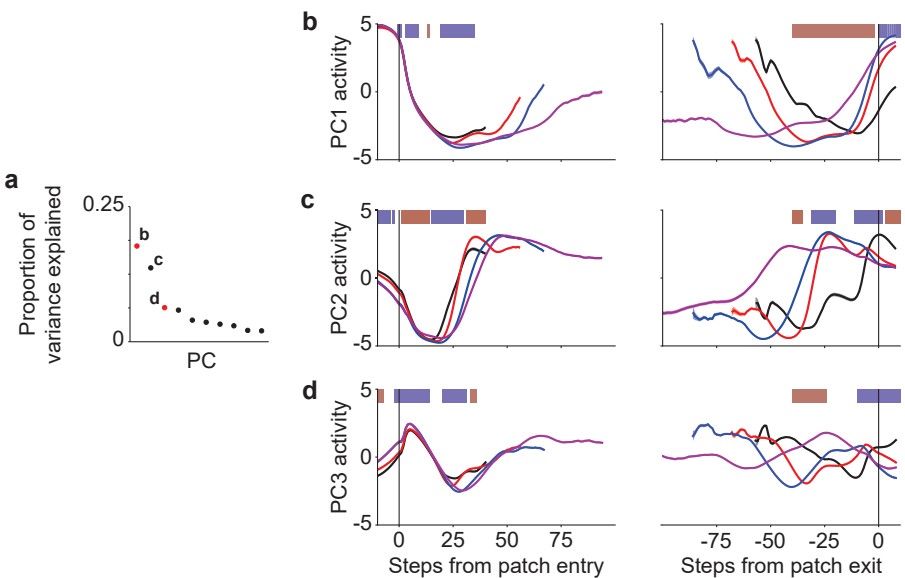

Figure A.2: Major principal components from the LSTM layer of a representative trained agent. **a)** Proportion of LSTM layer variance explained by each of the first 10 PCs. PC2 (c) is highlighted in the main text. **b, c, d)** Activation of the principal components accounting for the most amount of variance in the agent's LSTM layer dynamics. Average activation is aligned to patch entry (left), and patch exit (right). Shaded blue and red bars indicate steps where there is a significant slope-patch leaving time quartile relationship (negative in blue, positive in red). Traces in each quartile are plotted until median patch leaving time. Shaded regions along activity traces denote standard errors.

## A.2 Accounting for discounting in the marginal value theorem

We accounted for temporal discounting rate by simulating individual stay and leave decisions at many patch leaving steps. Agents could either stay for an additional step of reward before leaving a patch, or immediately leave the patch, where the subsequent 5000 steps were simulated as alternating between a fixed number of steps in a patch and a fixed number of steps traveling between patches. For example, on step 150 within a patch, and given a future fixed travel time of 50 steps and a future fixed patch time of 100 steps, is it more beneficial to leave immediately (Figure A.3a; "leave" in black) or stay in the patch for an additional step of reward (Figure A.3a; 150 in green)? By computing the discounted return for immediately leaving, and computing the discounted returns for remaining in the patch for one additional step of reward at every level of patch depletion, we generate a curve where the value of leaving can be compared to the value of staying at individual patch depletion points (Figure A.3b).

In Figure A.3b, we show the simulated indifference step when there are 100 fixed steps in a patch and 50 fixed steps traveling between patches. An indifference step can be estimated for every choice of subsequent fixed patch and travel steps (e.g., as in Figure A.3a), which gives rise to an indifference curve over these parameters (Figure A.3c). Using the observed mean travel steps from trained agents, we can instead sweep over only fixed steps in patch in order to evaluate an agent's choice of mean patch time. This assumes that the agent's mean travel time is constant and independent of mean patch time.

Where the stay/leave indifference curve matched the fixed patch steps (i.e., the unity line in Figure A.3c) provided an approximation of a single mean patch time that maximizes the discounted return (given a discount rate and observed mean travel steps).

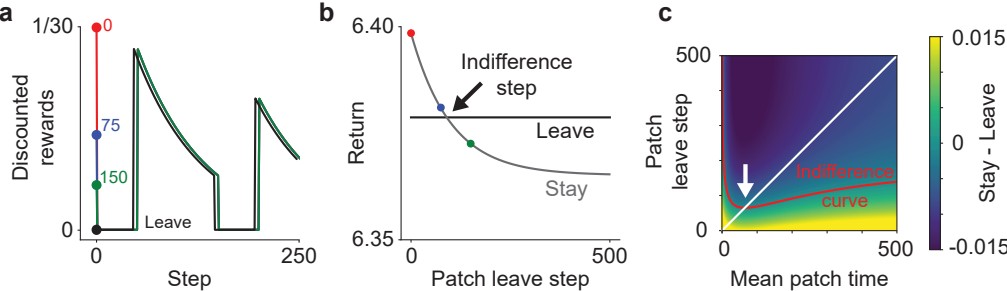

Figure A.3: Accounting for temporal discounting in the marginal value theorem. **a)** Simulated discounted rewards for different artificial policies. After the first step, these simulated agents alternate between a fixed number of steps in a patch, and a fixed number of steps traveling between patches. **b)** Computed discounted return for artificial policies in a). **c)** Difference in discounted returns between stay and leave policies at different fixed patch times (assuming a discount rate and a fixed travel time). White arrow indicates discounted MVT estimate for optimal mean patch time.

## A.3    Training details

We generally follow similar training procedures as for the models described in Cultural General Intelligence Team et al. (2022). Each agent was trained on an internal cluster for roughly 13 days, and used approximately 40 GiB RAM, 8 CPU, and 8 GPUs.

