# OpenReview forum: "Adaptive patch foraging in deep reinforcement learning agents"
_TMLR — Accepted by TMLR_

### Review · Reviewer_DAPD · 2022-12-19

**Summary Of Contributions:**

This paper conducts an empirical analysis of the behavior of deep reinforcement learning (RL) agents in an ecological patch foraging task. The main findings are that RL agents display behavioral patterns similar to biological agents, find optimal foraging strategies, and display neural network layer activities that are similar to non-human primate neural recordings.

**Audience:**

Yes

**Claims And Evidence:**

Yes

**Requested Changes:**

(Critical) By using the phrase "non-stationary equilibrium" on page 2, the authors seem to intend to convey some additional difficulty of the environment, but this is not true for the actual work done in this paper. I doubt the authors meant the word "equilibrium" in the sense of game theory with multiple agents, since this paper is only concerned with a single agent. "Non-stationary" does not accurately describe the actual concrete environment used in the paper, since time-dependence of the reward rate is not an issue because the agent observes the instantaneous reward rate. If the authors meant to say that a real-world patch foraging environment is a multi-agent problem where non-stationarity and equilibrium notions apply, then the authors ought to make it clear that these are additional difficulties that are not treated in the current paper.

(Minor) Given that there exists a known theoretical optimal strategy, it is not surprising that a deep RL agent can learn to find it. The paragraph on page 2 that stresses the difficulty of the environment appears excessive and can be toned down.

(Minor) The spatial trajectories shown in Figure 1d currently do not seem to serve much purpose in this paper. The authors should explain what a reader is supposed to learn from it.

**Strengths And Weaknesses:**

Strengths

This paper appears to be the first to highlight patch foraging as a potentially fruitful environment in which to study the behavior of deep RL agents. From the literature review, this environment contains ample theoretical and experimental biological research results to guide the behavioral investigation of deep RL agents.

The paper is very well organized and clearly written. The main claims and findings are stated precisely; the empirical results presented in figures are tightly connected to the claims they are meant to support; the results are rigorously analyzed and explained.

The findings of this paper is relevant and interesting to the broader AI community. I believe this paper falls under the category of "experimental and/or theoretical studies yielding new insight into the design and behavior of learning in intelligent systems" in the TMLR solicitation. While this paper rightly constrains its claims to the specific task of path foraging, one may venture to believe that it lends support to the broader hypothesis that deep RL agents can serve as low-fidelity surrogate models of real biological agents.


Weaknesses

In the environment used in this paper, as soon as an agent enters a patch, the other patch is instantaneously refreshed. This does not seem like a realistic assumption. While this paper merely adopts this assumption from the existing literature, it is still a limitation and weakness of this area of study.

Only one deep RL algorithm (MPO) is used for the entire behavioral analysis. This leaves the paper open to the criticism or concern that the findings may somehow be a fluke of MPO, that other deep RL algorithms could produce different conclusions.

The relation between this kind of behavioral research and the more mainstream agenda of designing better learning agents is not clear and should be discussed. One may not wish to replicate the behavior of biological agents if doing so inherits undesirable limitations of biological agents.

---

> ### Author Response · Authors · 2023-02-05
> **Response to Reviewer DAPD**
>
> We would like to thank the reviewer for their time, kind words, and comments on the paper. We have revised the text of the paper to address several of the points raised, and now feel the paper is stronger as a result of these changes.
>
> (1) “[...] as soon as an agent enters a patch, the other patch is instantaneously refreshed. This does not seem like a realistic assumption.”
>
> We agree that this environmental choice is not biologically realistic. However, this was chosen to adhere to the assumptions of the marginal value theorem (MVT; Charnov, 1976), which assumes that an agent’s choice of patch residence time does not impact the resources available outside of the current patch in which the agent is foraging. We have added new text, which discusses ecological realism for future experiments (almost all of which would lack a known optimal solution).
>
> (2) Performance of alternative models.
>
> This is a point common to all reviewers. We provide the same text below to all reviewers:
>
> We chose MPO as our algorithm of interest because in similar work, MPO agents were found to navigate even more complex environments successfully (Cultural General Intelligence Team, 2022). In this paper, we were focused on proof of possibility, and as such were focused on one kind of model. Although we agree that future work looking at the optimality of different RL methods on complex foraging tasks is an interesting avenue of research. We have now added the following clarification to the paper discussion:
> “While the current paper considers only the MPO reinforcement learning algorithm due to its success in a similar environment (Cultural General Intelligence Team, 2022), future work may find improvements in learning to solve the patch foraging problem with alternative methods.”
>
> (3) “The relation between this kind of behavioral research and the more mainstream agenda of designing better learning agents is not clear and should be discussed.”
>
> We now provide a broader impact statement that we hope satisfies some of the concerns about the applications of these results to artificial agent research.
>
> (4) A “non-stationary equilibrium”.
>
> Another reviewer also raised this point. We provide the same text below to relevant reviewers:
>
> We used this phrase when trying to communicate the difficulties in solving the patch foraging problem. Mainly, when agents learn to move more efficiently between patches, their travel time between patches decreases. The marginal value theorem then dictates that the optimal patch residence time also decreases. In all, agents could be performing optimally according to theory, but then improvements in the agent’s movements shift the optimal behavior. In this line of thinking, there is an optimal equilibrium between travel time and patch residence time. The problem is also non-stationary because the optimal patch residence time changes according to the agent’s own behavior, but also changes over time in real-world foraging conditions (e.g., moving from a plentiful to a scarce location, seasonal change in resources).
>
> Regardless, our use of the phrase seems to have been confusing or problematic, and so we have opted to change the text. It now reads:
> “Finally, changes in the resource-richness of the environment or the agent's movement time between patches impact the optimal patch time, as prescribed by the MVT.”
>
> (5) “The paragraph on page 2 that stresses the difficulty of the environment appears excessive and can be toned down.”
>
> We have edited the paragraph as requested, and hope the reviewer now finds the revised text acceptable.
>
> (6) “The spatial trajectories shown in Figure 1d currently do not seem to serve much purpose in this paper.”
>
> We disagree, and would like to keep the movement trajectories in the paper as-is. First, we find it useful to visually show the patch distance evaluation environments (6m, 8m, 10m, and 12m). On these panels, we can also show examples of agent movement trajectories without using any additional space. We think it is important to be transparent about the agent behavior given that the movement time between patches is a key part of the optimal solution in this problem.

---

> > ### Comment · Reviewer_DAPD · 2023-02-08
> > **Author response has clarified the confusion about equilibrium**
> >
> > The author's explanation about non-stationary equilibrium is helpful, and the removal of the phrase "non-stationary equilibrium" helps to avoid confusion. Most RL audiences are focused on global optimality (i.e., maximizing discounted return), so it's easy to fail to realize that the author is talking about local optimality of patch residence time, which as the author explains depends on the agent's current behavior.

---

### Review · Reviewer_1JDk · 2022-12-19

**Summary Of Contributions:**

The authors show that deep reinforcement learning is able to learn optimal patch foraging solutions in a simulated environment. The authors test their method on a simulated 3D foraging environment and compare the performance of their trained RL agent to a random baseline agent. These learned agents are able to skillfully trade off travel time and patch rewards. Furthermore, the internal representations of these agents are similar to those of biological agents.


**Audience:**

Yes

**Broader Impact Concerns:**

There do not appear to be any significant ethical concerns raised by the work presented in this paper. However, the authors should consider adding a Broader Impact Statement to address any potential implications of their work for the use of RL in real-world applications, particularly in situations where foraging behavior is important (e.g., resource management, environmental monitoring).


**Claims And Evidence:**

Yes

**Requested Changes:**

- The authors should provide a more detailed discussion of the limitations of their method and potential avenues for future work. This could include a discussion of the challenges of using model-free RL to solve the patch foraging problem, as well as potential alternative approaches that might be more successful.
- The authors could also provide a better baseline RL agent than just random and how it compares to the performance of the trained RL agent. Perhaps they could even compare to how humans accomplish this task? I think some literature already looks at this, so maybe just including that data to compare against would be interesting.
- The trained agent trajectories in figure 1 don’t look optimal. Why is it taking long paths between the two patches?
- “Patch leaving times, movement policies, and changes in the resource-richness of the environment additionally impact the average reward rate of the environment for an agent, which can create a non-stationary equilibrium.
    - I'm a bit confused by this sentence. What is meant by a non-stationary equilibrium? I would think this is just an MDP problem, so I don’t see what is non-stationary, and what dynamics are in equilibrium?
- It would be helpful if the authors provided more context for the human performance data, including details about the sample size and any other relevant characteristics of the participants.


**Strengths And Weaknesses:**

Overall, I think this paper provides an interesting connection between cognitive science/biology and reinforcement learning. There is a large literature modeling biological behaviors with reinforcement learning, and patch foraging seems well-studied, so I believe that the respective communities will be interested in this paper. One strength of this paper is that the authors clearly articulate the motivation for their work and provide a thorough explanation of the MVT and how it relates to patch foraging behavior in biological systems. The authors also present a detailed description of their experimental setup and method, making it easy for readers to understand how the RL agents were trained and evaluated.

One weakness of the paper is that the authors do not adequately discuss the limitations of their method or how it might be improved in future work. For example, the authors mention that the optimal patch foraging solution may be difficult to discover using model-free RL, but they do not explore why this might be the case or what alternative approaches might be more successful. In fact, it would be more surprising to me if deep RL with an LSTM wouldn’t be able to learn patch foraging, given recent successes of deep RL on simulated games.

---

> ### Author Response · Authors · 2023-02-05
> **Response to Reviewer 1JDk**
>
> We would like to thank the reviewer for their time and insightful comments on the paper. We have revised the text of the paper to address several of the points raised.
>
> (1) “The authors should provide a more detailed discussion of the limitations of their method and potential avenues for future work.”
>
> In the revised text, we now discuss additional limitations and future directions. These include our choice of the MPO algorithm, and how future foraging environments may be made more biologically realistic.
>
> (2) Performance of alternative models.
>
> This is a point common to all reviewers. We provide the same text below to all reviewers:
>
> We chose MPO as our algorithm of interest because in similar work, MPO agents were found to navigate even more complex environments successfully (Cultural General Intelligence Team, 2022). In this paper, we were focused on proof of possibility, and as such were focused on one kind of model. Although we agree that future work looking at the optimality of different RL methods on complex foraging tasks is an interesting avenue of research. We have now added the following clarification to the paper discussion:
> “While the current paper considers only the MPO reinforcement learning algorithm due to its success in a similar environment (Cultural General Intelligence Team, 2022), future work may find improvements in learning to solve the patch foraging problem with alternative methods.”
>
> (3) “The trained agent trajectories in figure 1 don’t look optimal.”
>
> We agree—agent movement after training does not follow an expected straight line between the two patches. We believe more could be done for the movement performance of these agents, and now discuss it briefly in the revised text.
>
> Regardless, given the time the agents take to (sub-optimally) travel between the two patches, they interestingly stay within patches for the optimal amount of time given their movement time and discount rates. According to the MVT, when agents improve their movement policy, the optimal patch residence time should decrease. Given this relationship, it may be interesting for future work to look at this tug of war between decreasing the movement time between patches and adapting patch residence time accordingly as learning progresses.
>
> (4) A “non-stationary equilibrium”.
>
> Another reviewer also raised this point. We provide the same text below to relevant reviewers:
>
> We used this phrase when trying to communicate the difficulties in solving the patch foraging problem. Mainly, when agents learn to move more efficiently between patches, their travel time between patches decreases. The marginal value theorem then dictates that the optimal patch residence time also decreases. In all, agents could be performing optimally according to theory, but then improvements in the agent’s movements shift the optimal behavior. In this line of thinking, there is an optimal equilibrium between travel time and patch residence time. The problem is also non-stationary because the optimal patch residence time changes according to the agent’s own behavior, but also changes over time in real-world foraging conditions (e.g., moving from a plentiful to a scarce location, seasonal change in resources).
>
> Regardless, our use of the phrase seems to have been confusing or problematic, and so we have opted to change the text. It now reads:
> “Finally, changes in the resource-richness of the environment or the agent's movement time between patches impact the optimal patch time, as prescribed by the MVT.”
>
> (5) “It would be helpful if the authors provided more context for the human performance data, including details about the sample size and any other relevant characteristics of the participants.”
>
> We are a bit confused about this particular point. Did the reviewer mean the sample size and demographics of human foraging experiments we have cited in the paper (e.g., Constantino & Daw, 2015; Pacheco-Cobos et al., 2019)? We did not collect any human data for this study.
>
> (6) Adding a Broader Impact Statement.
>
> We would like to thank the reviewer for this suggestion. We have now added a broader impact statement to the paper.

---

### Review · Reviewer_hyzv · 2023-01-24

**Summary Of Contributions:**

This paper studies the use of deep reinforcement learning in simulating patch foraging behavior in artificial intelligence. The study found that the agents were able to learn adaptive patterns similar to those seen in biological foragers and approached optimal behavior when accounting for temporal discounting. Additionally, the authors observed internal dynamics in the agents that resembled neural recordings from non-human primates. The study suggests that both biological and artificial agents may arrive at similar solutions when faced with similar ecological pressures, indicating a common foundation for adaptive, intelligent behavior.


**Audience:**

Yes

**Claims And Evidence:**

Yes

**Requested Changes:**

Questions:
* Can you provide a comparison with other existing methods in the literature to demonstrate the effectiveness of the proposed method?
* It would be better for the authors to provide more information about the ecological complexity of the environment used in the experiments, especially for an audience that is not familiar with patch foraging or biological intelligence.

**Strengths And Weaknesses:**

Strengths:
* The use of a 3D patch foraging environment inspired by experiments from behavioral ecology, which adds ecological complexity and realism to the task.
* The demonstration that the agents can learn to adapt their behavior in a way that is similar to biological foragers and approach optimal patch foraging behavior when accounting for temporal discounting.

Weaknesses:
* The paper does not provide a detailed description of the agents' architectures and hyperparameters used in the experiments.
* The paper does not provide a comparison with other existing methods in the literature to demonstrate the effectiveness of the proposed method.
* From the perspective of RL, (I guess) it is possible that the environment used in the experiments may not be complex enough to fully capture the real-world patch foraging phenomenon.

---

> ### Author Response · Authors · 2023-02-05
> **Response to Reviewer hyzv**
>
> We would like to thank the reviewer for their time and suggestions.
>
> (1) “The paper does not provide a detailed description of the agents' architectures and hyperparameters used in the experiments.”
>
> We apologize for any confusion. We have now restated in the paper that the model is the same as the one implemented in Cultural General Intelligence Team et al. (2022), and state in the paper where any changes have been made.
>
> (2) “The paper does not provide a comparison with other existing methods [...]”
>
> This is a point common to all reviewers. We provide the same text below to all reviewers:
>
> We chose MPO as our algorithm of interest because in similar work, MPO agents were found to navigate even more complex environments successfully (Cultural General Intelligence Team, 2022). In this paper, we were focused on proof of possibility, and as such were focused on one kind of model. Although we agree that future work looking at the optimality of different RL methods on complex foraging tasks is an interesting avenue of research. We have now added the following clarification to the paper discussion:
> “While the current paper considers only the MPO reinforcement learning algorithm due to its success in a similar environment (Cultural General Intelligence Team, 2022), future work may find improvements in learning to solve the patch foraging problem with alternative methods.”
>
> (3) Elaborating on the ecological complexity of the environment.
>
> Thank you for this suggestion. In the revised text, we have elaborated on the similarities and differences between our simulated patch foraging task and biological patch foraging. We have also elaborated on some directions where implementing more ecological realism may be of particular interest to future work.

---

### Decision · Action_Editors · 2023-03-14

**Recommendation:** Accept as is

**Comment:**

All of the reviewers recognized that this work is interesting and technically valid. They had some concerns about clarity, the lack of comparisons to other models, and some of the choice of words. But, they were largely satisfied with the revisions the authors made to address these concerns, and all recommended acceptance.

**Audience:**

This paper would be of interest to anyone in the machine learning field who is interested in more biologically realistic tasks, optimal decision making, and/or the connections between deep learning and neuroscience.

**Claims And Evidence:**

This paper claims that deep RL agents can learn to engage in patch foraging, will approach the optimal solution depending on the temporal discounting factor used, and will develop representations that share some properties with the representations seen in the brains of animals engaged in patch foraging. These claims are solidly backed up by empirical evidence of the behaviour and representations of MPO agents.